# Online Learning via the Differential Privacy Lens

**Jacob Abernethy**[*]
College of Computing
Georgia Institute of Technology
prof@gatech.edu

**Young Hun Jung**[*]
Department of Statistics
University of Michigan
yhjung@umich.edu

**Chansoo Lee**[*]
Google Brain
chansoo@google.com

**Audra McMillan**[*]
Simons Inst. for the Theory of Computing
Department of Computer Science
Boston University
Khoury College of Computer Sciences
Northeastern University
audramarymcmillan@gmail.com

**Ambuj Tewari**[*]
Department of Statistics
Department of EECS
University of Michigan
tewaria@umich.edu

## Abstract

In this paper, we use differential privacy as a lens to examine online learning in both full and partial information settings. The differential privacy framework is, at heart, less about privacy and more about algorithmic stability, and thus has found application in domains well beyond those where information security is central. Here we develop an algorithmic property called *one-step differential stability* which facilitates a more refined regret analysis for online learning methods. We show that tools from the differential privacy literature can yield regret bounds for many interesting online learning problems including online convex optimization and online linear optimization. Our stability notion is particularly well-suited for deriving first-order regret bounds for follow-the-perturbed-leader algorithms, something that all previous analyses have struggled to achieve. We also generalize the standard max-divergence to obtain a broader class called *Tsallis max-divergences*. These define stronger notions of stability that are useful in deriving bounds in partial information settings such as multi-armed bandits and bandits with experts.

## 1 Introduction

Stability of output in presence of small changes to input is a desirable feature of methods in statistics and machine learning [11, 19, 31, 42]. Another area of research for which stability is a core component is *differential privacy* (DP). As Dwork and Roth [15] observed, "differential privacy is enabled by stability and ensures stability." They argue that the "differential privacy lens" offers a fresh perspective to examine areas other than privacy. For example, the DP lens has been used successfully in designing coalition-proof mechanisms [25] and preventing false discovery in statistical analysis [10, 13, 17, 28].

In this paper, we use the DP lens to design and analyze randomized online learning algorithms in a variety of canonical online learning problems. The DP lens allows us to analyze a broad class of online learning problems, spanning both full information (online convex optimization (OCO), online linear optimization (OLO), experts problem) and partial information settings (multi-armed bandits, bandits with experts) using a unified framework. We are able to analyze follow-the-perturbed-leader (FTPL) as well as follow-the-regularized leader (FTRL) based algorithms resulting in both *zero-order* and

---

[*]Author order is alphabetical denoting equal contributions.

*first-order regret bounds*; see Section 2 for definitions of these bounds. However, our techniques are especially well-suited to proving first-order bounds for perturbation based methods. Historically, the understanding of the regularization based algorithms has been more advanced thanks to connections with ideas from optimization such as duality and Bregman divergences. There has been recent work [1] on developing a general framework to analyze FTPL algorithms, but it only yields zero-order bounds. A general framework that can yield first-order bounds for FTPL algorithms has been lacking so far, but we believe that the framework outlined in this paper may fill this gap in the literature. Our rich set of examples suggests that our framework will be useful in translating results from the DP literature to study a much larger variety of online learning problems in the future. This means that we immediately benefit from research advances in the DP community.

We emphasize that our aim is *not* to design low-regret algorithms that satisfy the privacy condition– there is already substantial existing work along these lines [4, 21, 39, 40]. Our goal is instead to show that, in and of itself, a DP-inspired stability-based methodology is quite well-suited to designing online learning algorithms with excellent guarantees. In fact, there are theoretical reasons to believe this should be possible. Alon et al. [7] have shown that if a class of functions is privately learnable, then it has finite Littlestone dimension (a parameter that characterizes learnability for online binary classification) via non-constructive arguments. Our results can be interpreted as proving analogous claims in a constructive fashion albeit for different, more tractable online learning problems.

In many of our problem settings, we are able to show new algorithms that achieve optimal or near-optimal regret. Although many of these regret bounds have already appeared in the literature, we note that they were previously possible only via specialized algorithms and analyses. In some cases (such as OLO), the regret bound itself is new. Our main technical contributions are as follows:

- We define *one-step differential stability* (Definitions 2.1 and 2.2) and derive a key lemma showing how it can yield first-order regret bounds (Lemma 3.1).
- *New algorithms with first-order bounds for both OCO* (Theorem 3.2) *and OLO problems* (Corollary 3.3) based on the objective perturbation method from the DP literature [23]. The OLO first-order bound is the first of its kind to the best of our knowledge.
- We introduce a novel family of *Tsallis $\gamma$-max-divergences* (see (2)) as a way to ensure tighter stability as compared to the standard max-divergence. Having tighter control on stability is crucial in partial information settings where loss estimates can take large values.
- We provide *optimal first-order bounds for the experts problem via new FTPL algorithms* using a variety of perturbations (Theorem 3.6).
- Our *unified analysis of multi-armed bandit algorithms* (Theorem 4.2) not only unifies the treatment of a large number of perturbations and regularizers that have been used in the past but also reveals the exact type of differential stability induced by them.
- *New perturbation-based algorithms for the bandits with experts problem* that achieve the same zero-order and first-order bounds (Theorem 4.3) as the celebrated EXP4 algorithm [9].

## 2 Preliminaries

The $\ell_\infty$, $\ell_2$, and $\ell_1$ norms are denoted by $\|\cdot\|_\infty$, $\|\cdot\|_2$, and $\|\cdot\|_1$ respectively. The vector $\mathbf{e}_i$ denotes the $i$th standard basis vector. The norm of a set $\mathcal{X}$ is defined as $\|\mathcal{X}\| = \sup_{x \in \mathcal{X}} \|x\|$. A sequence $(a_1, \ldots, a_t)$ is abbreviated $a_{1:t}$, and a set $\{1, \ldots, N\}$ is abbreviated $[N]$. For a symmetric matrix $S$, $\lambda_{\max}(S)$ denotes its largest eigenvalue. The probability simplex in $\mathbb{R}^N$ is denoted by $\Delta^{N-1}$. Full versions of omitted/sketched proofs can be found in the appendix.

### 2.1 Online learning

We adopt the common perspective of viewing online learning as a repeated game between a learner and an adversary. We consider an *oblivious adversary* that chooses a sequence of loss functions $\ell_t \in \mathcal{Y}$ before the game begins. At every round $t$, the learner chooses a move $x_t \in \mathcal{X}$ and suffers loss $\ell_t(x_t)$. The action spaces $\mathcal{X}$ and $\mathcal{Y}$ will characterize the online learning problem. For example, in multi-armed bandits, $\mathcal{X} = [N]$ for some $N$ and $\mathcal{Y} = [0,1]^N$. Note that the learner is allowed to access a private source of randomness in selecting $x_t$. The learner's goal is to minimize the *expected regret* after $T$ rounds:

$$\mathbb{E}[\mathsf{Regret}_T] = \mathbb{E}\left[\sum_{t=1}^T \ell_t(x_t)\right] - \min_{x \in \mathcal{X}} \sum_{t=1}^T \ell_t(x),$$

where the expectations are over all of the learner's randomness, and we recall that here the $\ell_t$'s are non-random. The minimax regret is given by $\min_{\mathcal{A}} \max_{\ell_{1:T}} \mathbb{E}[\text{Regret}_T]$ where $\mathcal{A}$ ranges over all learning algorithms. If an algorithm achieves expected regret within a constant factor of the minimax regret, we call it *minimax optimal* (or simply *optimal*). If the factor involved is not constant but logarithmic in $T$ and other relevant problem parameters, we call the algorithm *minimax near-optimal* (or simply *near-optimal*).

In the *loss/gain setting*, losses can be positive or negative. In the *loss-only setting*, losses are always positive: $\min_{x \in \mathcal{X}} \ell_t(x) \geq 0$ for all $t$. *Zero-order regret bounds* involve $T$, the total duration of the game. In the loss-only setting, a natural notion of the hardness of the adversary sequence is the cumulative loss of the best action in hindsight, $L_T^* = \min_{x \in \mathcal{X}} \sum_{t=1}^{T} \ell_t(x)$. Note that $L_T^*$ is uniquely defined even though the best action in hindsight (denoted by $x_T^*$) may not be unique. Bounds that depend on $L_T^*$ instead of $T$ adapt to the hardness of the actual losses encountered and are called *first-order regret bounds*. We will ignore factors logarithmic in $T$ when calling a bound first-order.

There are some special cases of online learning that arise frequently enough to have received names. In *online convex (resp. linear) optimization*, the functions $\ell_t$ are convex (resp. linear) and the learner's action set $\mathcal{X}$ is a subset of some Euclidean space. In the linear setting, we identify a linear function with its vector representation and write $\ell_t(x) = \langle \ell_t, x \rangle$. In the *experts problem*, we have $\mathcal{X} = [N]$ and we use $i_t$ instead of $x_t$ to denote the learner's moves. Also we write $\ell_t(i) = \ell_{t,i}$ for $i \in [N]$.

We consider both full and partial information settings. In the *full information setting*, the learner observes the loss function $\ell_t$ at the end of each round. In the *partial information setting*, learners receive less feedback. A common partial information feedback is *bandit feedback*, i.e., the learner only observes its own loss $\ell_t(x_t)$. Due to less amount of information available to the learner, deriving regret bounds, especially first-order bounds, is more challenging in partial information settings.

Note that, in settings where the losses are linear, we will often use $L_t = \sum_{s=1}^{t} \ell_s$ to denote the cumulative loss vector. In these settings, we caution the reader to distinguish between $L_T$, the final cumulative loss vector and the scalar quantity $L_T^*$.

## 2.2 Stability notions motivated by differential privacy

There is a substantial literature on stability-based analysis of statistical learning algorithms. However, there is less work on identifying stability conditions that lead to low regret online algorithms. A few papers that attempt to connect stability and online learning are interestingly all unpublished and only available as preprints [32, 35, 36]. To the best of our knowledge no existing work provides a stability condition which has an explicit connection to differential privacy and which is strong enough to use in both full information and partial information settings.

Differential privacy (DP) was introduced to study data analysis mechanism that do not reveal too much information about any single instance in a database. In this paper, we will use DP primarily as a stability notion [15, Sec. 13.2]. DP uses the following divergence to quantify stability. Let $P, Q$ be distributions over some probability space. The *$\delta$-approximate max-divergence* between $P$ and $Q$ is

$$D_\infty^\delta(P, Q) = \sup_{P(B) > \delta} \log \frac{P(B) - \delta}{Q(B)}, \tag{1}$$

where the supremum is taken over measurable sets $B$. When $\delta = 0$, we drop the superscript $\delta$. If $Y$ and $Z$ are random variables, then $D_\infty^\delta(Y, Z)$ is defined to be $D_\infty^\delta(P_Y, P_Z)$ where $P_Y$ and $P_Z$ are the distributions of $Y$ and $Z$. We want to point out that the max-divergence is *not* a metric, because it is asymmetric and does not satisfy the triangle inequality.

A randomized online learning algorithm maps the loss sequence $\ell_{1:t-1} \in \mathcal{Y}^{t-1}$ to a distribution over $\mathcal{X}$. We now define a stability notion for online learning algorithms that quantifies how much does the distribution of the algorithm change when a new loss function is seen.

**Definition 2.1** (One-step differential stability w.r.t. a divergence)**.** *An online learning algorithm $\mathcal{A}$ is one-step differentially stable w.r.t. a divergence $D$ (abbreviated DiffStable(D)) at level $\epsilon$ iff for any $t$ and any $\ell_{1:t} \in \mathcal{Y}^t$, we have $D(\mathcal{A}(\ell_{1:t-1}), \mathcal{A}(\ell_{1:t})) \leq \epsilon$.*

**Remark.** *The classical definition of DP [4] says a randomized algorithm $\mathcal{A}$ is $(\epsilon, \delta)$-DP if $D_\infty^\delta(\mathcal{A}(\ell_{1:t}), \mathcal{A}(\ell'_{1:t})) \leq \epsilon$ whenever $\ell_{1:t}$ and $\ell'_{1:t}$ differ by at most one item. We can consider $\ell_{1:t-1}$ as $\ell'_{1:t}$ by adding a uninformative loss (e.g., zero loss for any action) in the last item.*

In the case when $\ell_t$ is a vector, it will be useful to define a similar notion where the stability level depends on the norm of the last loss vector.

**Definition 2.2** (One-step differential stability w.r.t. a norm and a divergence). *An online learning algorithm $\mathcal{A}$ is one-step differentially stable w.r.t. a norm $\|\cdot\|$ and a divergence $D$ (abbreviated DiffStable(D,$\|\cdot\|$)) at level $\epsilon$ iff for any $t$ and any $\ell_{1:t} \in \mathcal{Y}^t$, we have $D(\mathcal{A}(\ell_{1:t-1}), \mathcal{A}(\ell_{1:t})) \leq \epsilon\|\ell_t\|$.*

As we will discuss later, in the partial information setting, the estimated loss vectors can have very large norms. In such cases, it will be helpful to consider divergences that give a tighter control compared to the max-divergence. Define a new divergence (we call it *Tsallis $\gamma$-max-divergence*)

$$D_{\infty,\gamma}(P,Q) = \sup_B \log_\gamma(P(B)) - \log_\gamma(Q(B)), \tag{2}$$

where the generalized logarithm $\log_\gamma$, which we call Tsallis $\gamma$-logarithm[2] is defined for $x \geq 0$ as

$$\log_\gamma(x) = \begin{cases} \log(x) & \text{if } \gamma = 1 \\ \frac{x^{1-\gamma}-1}{1-\gamma} & \text{if } \gamma \neq 1 \end{cases}.$$

For given $P$ and $Q$, $D_{\infty,\gamma}(P,Q)$ is a non-decreasing function of $\gamma$ for $\gamma \in [1,2]$ (see Appendix A). Therefore, $\gamma > 1$ gives notions *stronger* than standard differential privacy. We will only consider the case $\gamma \in [1,2]$ in this paper. For the full information setting $\gamma = 1$ (i.e., the standard max divergence) suffices. Higher values of $\gamma$ are used only in the partial information settings. While our work shows the importance of the Tsallis max-divergences in the analysis of online learning algorithms, whether they lead to interesting notions of privacy is less clear. Along these lines, note that the definition of these divergences ensures that they enjoy *post-processing inequality* under deterministic functions just like the standard max-divergence (see Appendix A). Our generalization of the max-divergence does not rest on the use of an approximation parameter $\delta$; hence we will either use $D_\infty^\delta$ or $D_{\infty,\gamma}$, but never $D_{\infty,\gamma}^\delta$. Note that we often omit $\delta$ and $\gamma$ in cases where the former is 0 and the latter is 1.

## 3 Full information setting

In this section, we state a key lemma connecting the differential stability to first-order bounds. The lemma is then applied to obtain first-order bounds for OCO and OLO. In Section 3.3, we consider the Gradient Based Prediction Algorithm (GBPA) for the experts problem. There we show how the Tsallis max-divergence arises in GBPA analysis: a *differentially consistent* potential leads to a one-step differentially stable algorithm w.r.t. the Tsallis max-divergence (Proposition 3.5). Differential consistency was introduced as a smoothness notion for GBPA potentials by Abernethy et al. [2]. Skipped proofs appear in Appendix B.

### 3.1 Key lemma

The following lemma is a key tool to derive first-order bounds. There are two reasons why this simple lemma is so powerful. First, it makes the substantial body of algorithmic work in DP available for the purpose of deriving regret bounds. The parameters $\epsilon, \delta$ below then come directly from whichever algorithm from the DP literature we decide to use. Second, DP algorithms often add perturbations to achieve privacy. In that case, the fictitious algorithm $\mathcal{A}^+$ becomes the so-called "be-the-perturbed-leader" (BTPL) algorithm [22] whose regret is usually independent of $T$ (but does scale with $\epsilon, \delta$). One can generally set $\delta$ to be very small, such as $O(1/(BT))$, without significantly sacrificing the stability level $\epsilon$.

In the following lemma we consider taking an algorithm $\mathcal{A}$ and modifying it into a *fictitious* algorithm $\mathcal{A}^+$. This new algorithm has the benefit of one-step lookahead: at time $t$ $\mathcal{A}^+$ plays the distribution $\mathcal{A}(\ell_{1:t})$, whereas $\mathcal{A}$ would play $\mathcal{A}(\ell_{1:t-1})$. It is convenient to consider the regret of $\mathcal{A}^+$ for the purpose of analysis.

**Lemma 3.1.** *Consider the loss-only setting with loss functions bounded by $B$. Let $\mathcal{A}$ be DiffStable($D_\infty^\delta$) at level $\epsilon \leq 1$. Then we have*

$$\mathbb{E}[\mathsf{Regret}(\mathcal{A})_T] \leq 2\epsilon L_T^* + 3\mathbb{E}[\mathsf{Regret}(\mathcal{A}^+)_T] + \delta BT.$$

**Algorithm 1** Online convex optimization using Obj-Pert by Kifer et al. [23]

---

1: **Parameters** Privacy parameters $(\epsilon, \delta)$, upper bound $\beta$ on norm of loss gradient, upper bound $\gamma$ on eigenvalues of loss Hessian, perturbation distribution either Gamma or Gaussian
2: **for** $t = 1, \cdots, T$ **do**
3:    **if** using the Gamma distribution **then**
4:       Sample $b \in \mathbb{R}^d$ from a distribution with density $f(b) \propto \exp(-\frac{\epsilon \|b\|_2}{2\beta})$
5:    **else if** using the Gaussian distribution **then**
6:       Sample $b \in \mathbb{R}^d$ from the multivariate Gaussian $\mathcal{N}(0, \Sigma)$ where $\Sigma = \frac{\beta^2 \log \frac{2}{\delta} + 4\epsilon}{\epsilon^2} I$
7:    **end if**
8:    Play $x_t = \arg\min_{x \in \mathcal{X}} \sum_{s=1}^{t-1} \ell_s(x) + \frac{\gamma}{\epsilon} \|x\|_2^2 + \langle b, x \rangle$
9: **end for**

---

## 3.2 Online optimization: convex and linear loss functions

We now consider online convex optimization, a canonical problem in online learning, and show how to apply a privacy-inspired stability argument to effortlessly convert differential privacy guarantees into online regret bounds. In the theorem below, we build on the privacy guarantees provided by Kifer et al. [23] for (batch) convex loss minimization, and therefore Algorithm 1 uses their Obj-Pert (objective perturbation) method to select moves.

**Theorem 3.2** (First-order regret in OCO). *Suppose we are in the loss-only OCO setting. Let $\mathcal{X} \subset \mathbb{R}^d$, $\|\mathcal{X}\|_2 \leq D$ and let all loss functions be bounded by $B$. Further assume that $\|\nabla \ell_t(x)\|_2 \leq \beta$, $\lambda_{\max}(\nabla^2 \ell_t(x)) \leq \gamma$ and that the Hessian matrix $\nabla^2 \ell_t(x)$ has rank at most one, for every $t$ and $x \in \mathcal{X}$. Then, the expected regret of Algorithm 1 is at most $O(\sqrt{L_T^*(\gamma D^2 + \beta d D)})$ and $O\left(\sqrt{L_T^*(\gamma D^2 + D\sqrt{d(\beta^2 \log(BT))})}\right)$ with Gamma and Gaussian perturbations, respectively.*

*Proof Sketch.* From the DP result by Kifer et al. [23, Theorem 2], we can infer that $D_\infty^\delta(x_t, x_{t+1}) \leq \epsilon$, where $\delta$ becomes zero when using the Gamma distribution. This means that Algorithm 1 enjoys the one-step differential stability w.r.t. $D_\infty$ (resp. $D_\infty^\delta$) in the Gamma (resp. Gaussian) case. The regret of the fictitious $\mathcal{A}^+$ algorithm can be shown to be bounded by $\frac{\gamma}{\epsilon} D^2 + 2D\mathbb{E}\|b\|_2$. Using Lemma 3.1, we can deduce that the expected regret of Algorithm 1 is at most

$$2\epsilon L_T^* + \frac{3\gamma}{\epsilon} D^2 + 6D\mathbb{E}\|b\|_2 + \delta BT, \tag{3}$$

where $\delta$ becomes zero when using the Gamma distribution. We have $\mathbb{E}\|b\|_2 = \frac{2d\beta}{\epsilon}$ in the Gamma case and $\mathbb{E}\|b\|_2 \leq \frac{\sqrt{d(\beta^2 \log \frac{2}{\delta} + 4\epsilon)}}{\epsilon}$ in the Gaussian case. Plugging these results in (3) and optimizing $\epsilon$ (setting $\delta = \frac{1}{BT}$ for the Gaussian case) prove the desired bound. $\square$

The rank-one restriction on the Hessian of $\ell_t$, which allows loss curvature in one dimension, is a strong assumption but indeed holds in many common scenarios, e.g., $\ell_t(x) = \phi_t(\langle x, z_t \rangle)$ for some scalar loss function $\phi_t$ and vector $z_t \in \mathbb{R}^d$. This is a common situation in online classification and online regression with linear predictors. Moreover, it seems likely that the rank restriction can be removed in the results of Kifer et al. [23] at the cost of a higher $\epsilon$. A key strength of our approach is that we will immediately inherit any future improvements to existing privacy results. Note that first-order bounds for smooth convex functions have been shown by Srebro et al. [37]. However, their analysis relies on the self bounding property, i.e., the norm of the loss gradient is bounded by the loss itself, which does not hold for linear functions. Even logarithmic rates in $L_T^*$ are available [29] but they rely on extra properties such as exp-concavity. When the functions are linear, $\ell_t(x) = \langle \ell_t, x \rangle$, the restrictions on the Hessian are automatically met. The gradient condition reduces to $\|\ell_t\|_2 \leq \beta$ which gives us the following corollary.

**Corollary 3.3** (First-order regret in OLO). *Suppose we are in the loss-only OLO setting. Let $\mathcal{X} \subset \mathbb{R}^d$, $\|\mathcal{X}\|_2 \leq D$. Further assume that $\|\ell_t\|_2 \leq \beta$, for every $t$. Then, the expected regret of Algorithm 1 with no $\ell_2$-regularization (i.e., $\gamma = 0$) is at most $O(\sqrt{L_T^* d\beta D})$ and $O(\sqrt{L_T^* \beta D \sqrt{d \log(\beta DT)}})$ with Gamma and Gaussian perturbations, respectively.*

---

**Algorithm 2** Gradient-Based Prediction Algorithm (GBPA) for experts problem

---
1: **Input:** Concave potential $\tilde{\Phi} : \mathbb{R}^N \to \mathbb{R}$ with $\nabla \tilde{\Phi} \in \Delta^{N-1}$
2: Set $L_0 = 0 \in \mathbb{R}^N$
3: **for** $t = 1$ to $T$ **do**
4:     **Sampling:** Choose $i_t \in [N]$ according to distribution $p_t = \nabla \tilde{\Phi}(L_{t-1}) \in \Delta^{N-1}$
5:     **Loss:** Incur loss $\ell_{t,i_t}$ and observe the entire vector $\ell_t$
6:     **Update:** $L_t = L_{t-1} + \ell_t$
7: **end for**

---

Abernethy et al. [1] showed that FTPL with Gaussian perturbations is an algorithm applicable to general OLO problems with regret $O(\beta D \sqrt[4]{d} \sqrt{T})$. However, their analysis technique based on convex duality does not lead to first-order bounds in the loss-only setting. To the best of our knowledge, the result above provides a novel first-order bound for OLO when both the learner and adversary sets are measured in $\ell_2$ norm (the classic FTPL analysis of Kalai and Vempala [22] for OLO uses $\ell_1$ norm). Note that $L_T^*$ can be significantly less than its maximum value $\beta D T$. We also emphasize that the bound in Corollary 3.3 depends on the dimension $d$, which can lead to a loose bound. There are different algorithms such as online gradient descent or online mirror descent (e.g., see [20]) whose regret bound is dimension-free. It remains as an open question to prove such a dimension-free bound for any FTPL algorithm for OLO.

## 3.3 Experts problem

We will now turn our attention to another classical online learning setting of *prediction with expert advice* [12, 18, 24]. In the experts problem, $\mathcal{X} = [N]$, $\mathcal{Y} = [0,1]^N$ and a randomized algorithm plays a distribution over the $N$ experts. In the remainder of this paper, we will consider discrete sets for the players moves and so we will use $i_t$ instead of $x_t$ to denote the learner's move and $p_t \in \Delta^{N-1}$ to denote the distribution from which $i_t$ is sampled.

The GBPA family of algorithms is important for the experts problem and for the related problem of adversarial bandits (discussed in the next section). It includes as subfamilies FTPL and FTRL algorithms. The main ingredient in GBPA is a potential function $\tilde{\Phi}$ whose gradient is used to generate probability distributions for the moves. This potential function can be thought of as a *smoothed* version of the baseline potential $\Phi(L) = \min_i L_i$ for $L \in \mathbb{R}^N$. The baseline potential is non-smooth and using it in GBPA would result in the follow-the-leader (FTL) algorithm which is known to be unstable. FTRL and FTPL can be viewed as two distinct ways of smoothing the underlying non-smooth potential. In particular, FTPL uses *stochastic smoothing* by considering $\tilde{\Phi}$ of the form $\tilde{\Phi}_{\mathcal{D}}(L) = \mathbb{E}[\min_i(L_i - Z_i)]$ where $Z_i$'s are $N$ i.i.d. draws from the distribution $\mathcal{D}$. FTRL uses smoothed potentials of the form $\tilde{\Phi}_F(L) = \min_p (\langle p, L \rangle + F(p))$ for some strictly convex $F$.

### 3.3.1 From differential consistency to one-step differential stability

Abernethy et al. [2] analyzed the GBPA by introducing *differential consistency* defined below. The definition differs slightly from the original because it is formulated here with losses instead of gains. The notations $\nabla_{ii}^2$ and $\nabla_i$ are used to refer to specific entries in the Hessian and gradient respectively.

**Definition 3.4** (Differential consistency)**.** *We say that a function $f : \mathbb{R}^N \to \mathbb{R}$ is $(\gamma, \epsilon)$-differentially consistent if $f$ is twice-differentiable and $-\nabla_{ii}^2 f \leq \epsilon (\nabla_i f)^\gamma$ for all $i \in [N]$.*

This functions as a new measure of the potential's smoothness. Their main idea is to decompose the regret into three penalties [2, Lemma 2.1] and bound one of them when the potential function is differentially consistent. In fact, it can be shown that the potentials in many FTPL and FTRL algorithms are differentially consistent, and this observation leads to regret bounds of such algorithms.

Quite surprisingly, we can establish the one-step stability when the algorithm is the GBPA with a differentially consistent potential function. To state the proposition, we need to introduce a technical definition. We say that a matrix is *positive off-diagonal* (POD) if its off-diagonal entries are non-negative and its diagonal entries are non-positive. In the FTPL case where $F(p, Z) = -\langle p, Z \rangle$, it was already shown by Abernethy et al. [1] that $-\nabla^2 \tilde{\Phi}(L)$ is POD. It is easy to show that if $F(p) = $

$\sum_i f(p_i)$ for a strictly convex and smooth $f$, then $-\nabla^2 \tilde{\Phi}(L)$ is always POD (see Appendix B). The next proposition connects differential consistency to the one-step differential stability.

**Proposition 3.5** (Differential consistency implies one-step differential stability). *Suppose $\tilde{\Phi}(L)$ is of the form $\mathbb{E}[\min_p \langle L, p \rangle + F(p, Z)]$ and $\gamma \geq 1$. If $\tilde{\Phi}$ is $(\gamma, \epsilon)$-differentially consistent and $-\nabla^2 \tilde{\Phi}$ is always POD, the GBPA using $\tilde{\Phi}$ as potential is DiffStable($D_{\infty, \gamma}, \|\cdot\|_\infty$) at level $2\epsilon$.*

### 3.3.2 Optimal family of FTPL algorithms

We leverage our result from the previous section to prove that FTPL algorithms with a variety of perturbations have the minimax optimal first-order regret bound in the experts problem.

**Theorem 3.6** (First-order bound for experts via FTPL). *For the loss-only experts setting, FTPL with Gamma, Gumbel, Fréchet , Weibull, and Pareto perturbations, with a proper choice of distribution parameters, all achieve the optimal $O(\sqrt{L_T^* \log N} + \log N)$ expected regret.*

Although the result above is not the first optimal first-order bound for the experts problem, such a bound for FTPL with the wide variety of distributions mentioned above is not found in the literature. Previous FTPL analysis achieving first-order regret bounds all relied on specific choices such as exponential [22] and dropout [41] perturbations. There are results that consider Gaussian pertubations [1], random-walk perturbations [14], and a large family of symmetric distributions [34], but they only provide zero-order bounds.

## 4 Partial information setting

In this section, we provide stability based analyses of extensions of the GBPA framework to $N$-armed bandits and $K$-armed bandits with $N$ experts. All omitted proofs can be found in Appendix C.

### 4.1 GBPA for multi-armed bandits

In the multi-armed bandit setting, only the loss $\ell_{t, i_t}$ of the algorithm's chosen action is revealed. The GBPA for this setting is almost same as Algorithm 2 but with an extra step of loss estimation. The algorithm uses importance weighting to produce estimates $\hat{\ell}_t = \frac{\ell_{t, i_t}}{p_{t, i_t}} \mathbf{e}_{i_t}$ of the actual loss vectors—these are unbiased as long as $p_t$ has full support—and feeds these estimates to the standard GBPA using a (smooth) potential $\tilde{\Phi}$. Algorithm 4 in Appendix C summarizes these steps.

The losses fed to the full information GBPA are scaled by $1/p_{t, i_t}$ which can be very large. On the other hand, there is a special structure in $\hat{\ell}_t$: it has at most one non-zero entry. The following lemma is a replacement for the key lemma (Lemma 3.1) that exploits this special structure. The first term in the bound is the analogue of the $2\epsilon L_T^*$ term in the key lemma. This term shows that using $D_{\infty, \gamma}$ instead of using $D_\infty$ to measure stability can be useful in the bandit setting: the larger $\gamma$ is, the less problem we have from the inverse probability weighting inherent in $\hat{\ell}_{t, i_t}$. The second term in the bound is the analogue of the loss of the fictitious algorithm which now depends on the (expected) range of values attained by the (possibly random) function $F(p, Z)$.

**Lemma 4.1.** *Suppose the full information GBPA uses a potential of the form $\tilde{\Phi}(L) = \mathbb{E}[\min_p \langle L, p \rangle + F(p, Z)]$ and $\gamma \in [1, 2]$. If the full information GBPA is DiffStable($D_{\infty, \gamma}, \|\cdot\|_\infty$) at level $\epsilon$, then the expected regret of Algorithm 4 (in Appendix C) can be bounded as*

$$\mathbb{E}\left[\sum_{t=1}^T \ell_{t, i_t}\right] - L_T^* \leq \epsilon \, \mathbb{E}\left[\sum_{t=1}^T \hat{\ell}_{t, i_t}^2 p_{t, i_t}^\gamma\right] + \mathbb{E}\left[\max_p F(p, Z) - \min_p F(p, Z)\right].$$

We will now use this lemma to analyze a variety of FTPL and FTRL algorithms. Recall that an algorithm is in the FTPL family when $F(p, Z) = -\langle p, Z \rangle$, and is in the FTRL family when $F(p, Z) = F(p)$ for some deterministic regularization function $F(\cdot)$. There is a slight complication in the FTPL case: for a given $L$ computing the probability $p_{t, i} = \nabla_i \tilde{\Phi}_\mathcal{D}(L) = \mathbb{P}(i = \arg\min_{i'} (L_{i'} - Z_{i'}))$ is intractable even though we can easily draw samples from this probability distribution. A method called Geometric Resampling [27] solves this problem by computing a Monte-Carlo estimate of $1/p_{t, i}$ (which is all that is needed to run Algorithm 4). They show that the extra error due to this

---

**Algorithm 3** GBPA for bandits with experts problem

---
1: **Input:** Concave potential $\tilde{\Phi} : \mathbb{R}^N \to \mathbb{R}$, with $\nabla \tilde{\Phi} \in \Delta^{N-1}$, clipping threshold $0 \le \rho < 1/K$
2: Set $\overline{\phi}_0 = 0 \in \mathbb{R}^N$
3: **for** $t = 1$ to $T$ **do**
4:     **Probabilities over experts via gradient:** $p_t = \nabla \tilde{\Phi}(\overline{\phi}_{t-1}) \in \Delta^{N-1}$
5:     **Convert probabilities from experts to actions:** $q_t = \psi_t(p_t) = \sum_{j=1}^{K} \sum_{i:E_{i,t}=j} p_{t,i} \mathbf{e}_j \in \Delta^{K-1}$
6:     **Clipping (optional):** $\tilde{q}_t = C_\rho(q_t)$ where $C_\rho$ is defined in (4)
7:     **Sampling:** Choose $j_t \in [K]$ according to distribution $q_t$ (or $\tilde{q}_t$ if clipping)
8:     **Loss:** Incur loss $\ell_{t,j_t}$ and observe this value
9:     **Estimation:** Compute an estimate of loss vector $\hat{\ell}_t = \frac{\ell_{t,j_t}}{q_{t,j_t}} \mathbf{e}_{j_t} \in \mathbb{R}^K$ (or $\frac{\ell_{t,j_t}}{\tilde{q}_{t,j_t}} \mathbf{e}_{j_t}$ if clipping)
10:     **Convert estimate from actions to experts:** $\phi_t(\hat{\ell}_t) = \sum_{j=1}^{K} \sum_{i:E_{i,t}=j} \hat{\ell}_{t,j} \mathbf{e}_i \in \mathbb{R}^N$
11:     **Update:** $\overline{\phi}_t = \overline{\phi}_{t-1} + \phi_t(\hat{\ell}_t)$
12: **end for**

---

estimation is at most $KT/M$, where $M$ is the maximal number of samples per round that we use for the Monte-Carlo simulation. This implies that having $M = \Theta(\sqrt{T})$ for the zero-order bound or $M = \Theta(T)$ for the first-order bound would not affect the order of our bounds. Furthermore, they also prove that the expected number of samples to run the Geometric Resampling is constant per round (see [27, Theorem 2]). For simplicity, we will ignore this estimation step and assume the exact value of $p_{t,i_t}$ is available to the learner.

**Theorem 4.2** (Zero-order and first-order regret bounds for multi-armed bandits)**.** *Algorithm 4 (in Appendix C) enjoys the following bounds when used with different perturbations/regularizers:*

1. *FTPL with Gamma, Gumbel, Fréchet , Weibull, and Pareto pertubations (with a proper choice of distribution parameters) all achieve near-optimal expected regret of $O(\sqrt{NT \log N})$.*
2. *FTRL with Tsallis neg-entropy $F(p) = -\eta \sum_{i=1}^{N} p_i \log_\alpha(1/p_i)$ for $0 < \alpha < 1$ (with a proper choice of $\eta$) achieves optimal expected regret of $O(\sqrt{NT})$.*
3. *FTRL with log-barrier regularizer $F(p) = -\eta \sum_{i=1}^{N} \log p_i$ (with a proper choice of $\eta$) achieves expected regret of $O(\sqrt{NL_T^* \log(NT)} + N \log(NT))$.*

The proofs of the above results use the one-step differential stability as the unifying theme: in Part 1 we establish stability w.r.t. $D_\infty$, in Part 2, w.r.t. $D_{\infty,2-\alpha}$, and in Part 3, w.r.t. $D_{\infty,2}$. Parts 1-2 essentially rederive the results of Abernethy et al. [2] in the differential stability framework. Part 3 is quite interesting since it uses the strongest stability notion used in this paper (w.r.t. $D_{\infty,2}$). First-order regret bounds for multi-armed bandits have been obtained via specialized analysis several times [6, 26, 33, 38]. Such is the obscure nature of these analyses that in one case the authors claimed novelty without realizing that earlier first-order bounds existed! The intuition behind such analyses remained a bit unclear. Our analysis of the log-barrier regularizer clearly indicates why it enjoys first-order bounds (ignoring $\log(NT)$ term): the resulting full information algorithm enjoys a particularly strong form of one-step differential stability.

## 4.2 GBPA for bandits with experts

We believe that our unified differential stability based analysis of adversarial bandits can be extended to more complex partial information settings. We provide evidence for this by considering the problem of *adversarial bandits with experts*. In this more general problem, which was introduced in the same seminal work that introduced the adversarial bandits problem [9], there are $K$ actions and $N$ experts, $E_1, \ldots, E_N$, that at each round $t$, give advice on which of $K$ actions to take. The algorithm is supposed to combine their advice to pick a distribution $q_t \in \Delta^{K-1}$ and chooses an action $j_t \sim q_t$. Denote the suggestion of the $i$th expert at time $t$ as $E_{i,t} \in [K]$. Expected regret in this problem is defined as $\mathbb{E}\left[\sum_{t=1}^{T} \ell_{t,j_t}\right] - L_T^*$, where $L_T^*$ is now defined as $L_T^* = \min_{i=1}^{N} \sum_{t=1}^{T} \ell_{t,E_{i,t}}$.

The GBPA for this setting has a few more ingredients in it compared to the one for the multi-armed bandits. First, a transformation $\psi_t$ to convert $p_t \in \Delta^{N-1}$, a distribution over experts, to $q_t \in \Delta^{K-1}$, a distribution over actions: $\psi_t(p_t) = \sum_{j=1}^{K} \sum_{i:E_{i,t}=j} p_{t,i} \mathbf{e}_j$ , where $\mathbf{e}_j$ is the $j$th basis vector in $\mathbb{R}^K$. Note that the probability assigned to each action is the sum of the probabilities of all the experts that

recommended that action. Second, a transformation $\phi_t$ to convert the loss estimate $\hat{\ell}_t \in \mathbb{R}^K$ defined by $\hat{\ell}_{t,j_t} = \ell_{t,j_t}/q_{t,j_t}$ (and zero for $j \neq j_t$) into a loss estimate in $\mathbb{R}^N$: $\phi_t(\hat{\ell}_t) = \sum_{j=1}^K \sum_{i:E_{i,t}=j} \hat{\ell}_{t,j} \mathbf{e}_i$, where $\mathbf{e}_i$ is the $i$th basis vector in $\mathbb{R}^N$. At time $t$, the full information algorithm's output $p_t$ is used to select the action distribution $q_t = \psi_t(p_t)$ and the full information algorithm is fed $\phi_t(\hat{\ell}_t)$ to update $p_t$. Note that $\psi_t$ and $\phi_t$ are defined such that $\langle \psi_t(p), \hat{\ell} \rangle = \langle p, \phi_t(\hat{\ell}) \rangle$ for any $p \in \Delta^{N-1}$ and any $\hat{\ell} \in \mathbb{R}_+^K$. Lastly, the clipping function $C_\rho : \Delta^{K-1} \to \Delta^{K-1}$ is defined as:

$$[C_\rho(q)]_j = \begin{cases} \frac{q_j}{1-\sum_{j':q'_j < \rho} q_{j'}} & \text{if } q_j \geq \rho \\ 0 & \text{if } q_j < \rho \end{cases}. \tag{4}$$

It sets the probability weights that are less than $\rho$ to 0 and scales the rest to make it a distribution.

The clipping step (step 6) is optional. In fact, we can prove the zero-order bound without clipping, but this step becomes crucial to show the first-order bound. The main intuition is to bound the size of the loss estimate $\hat{\ell}_t$. With clipping, we can ensure $\|\hat{\ell}_t\|_\infty \leq 1/\rho$ for all $t$, which provides a better control on the one-step stability. The regret bounds in bandits with experts setting appear below.

**Theorem 4.3** (Zero-order and first-order regret bounds for bandits with experts)**.** *Algorithm 3 enjoys the following bounds when used with different perturbations such as Gamma, Gumbel, Fréchet , Weibull, and Pareto (with a proper choice of parameters).*

1. *With no clipping, it achieves near-optimal expected regret of $O(\sqrt{KT \log N})$.*
2. *With clipping, it achieves expected regret of $O\left((K \log N)^{1/3} (L_T^*)^{2/3}\right)$.*

The zero-order bound in Part 1 above was already shown for the celebrated EXP4 algorithm by Auer et al. [9]. Furthermore, Agarwal et al. [3] proved that, with clipping, EXP4 also enjoys a first-order bound with $O((L_T^*)^{2/3})$ dependence. Our theorem shows that EXP4 is not special in enjoying these bounds. The same bounds continue to hold for a variety of perturbation based algorithms. Such a result does not appear in the literature to the best of our knowledge. We note here that achieving $O(\sqrt{L_T^*})$ bounds in this setting was posed as on open problem by Agarwal et al. [3]. This problem was recently solved by the algorithm MYGA [5]. MYGA does achieve the optimal first-order bound, but the algorithm is not simple in that it has to maintain $\Theta(T)$ auxiliary experts in every round. In contrast, our algorithms are simple as they are all instances of GBPA along with the clipping idea.

**Acknowledgments**

Part of this work was done while AM was visiting the Simons Institute for the Theory of Computing. AM was supported by NSF grant CCF-1763786, a Sloan Foundation Research Award, and a postdoctoral fellowship from BU's Hariri Institute for Computing. AT and YJ were supported by NSF CAREER grant IIS-1452099. AT was also supported by a Sloan Research Fellowship. JA was supported by NSF CAREER grant IIS-1453304.

## Footnotes

[2]This quantity is often called the Tsallis $q$-logarithm, e.g., see [8, Chap. 4]

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
