[Supplementary Material]

# A    Proofs for Section 2

We provide the proofs of the claims made in Section 2.

## A.1    Tsallis $\gamma$-max-divergence gives tighter stability for larger $\gamma$

**Proposition A.1.** *Fix a distribution pair $P$ and $Q$. Then the function $D_{\infty,\gamma}(P,Q)$ is non-decreasing in $\gamma$ for $\gamma > 0$.*

*Proof.* Since $D_{\infty,\gamma}(P,Q) \geq 0$ (obvious by setting $B$ to be the entire space in (2)), we can fix a set $B$ with $P(B) \geq Q(B)$ and simply show that, for $0 < \gamma \leq \gamma'$,

$$\log_\gamma P(B) - \log_\gamma Q(B) \leq \log_{\gamma'} P(B) - \log_{\gamma'} Q(B).$$

This is equivalent to

$$\log_\gamma P(B) - \log_{\gamma'} P(B) \leq \log_\gamma Q(B) - \log_{\gamma'} Q(B).$$

Since $0 \leq Q(B) \leq P(B) \leq 1$, the above inequality will follow if we establish that the function

$$f(p) = \log_\gamma p - \log_{\gamma'} p$$

is non-increasing for $p \in (0,1]$. We can indeed verify this by taking the derivative

$$f'(p) = p^{-\gamma} - p^{-\gamma'},$$

which is non-positive since $p \leq 1$ and $\gamma' \geq \gamma > 0$. $\qquad\square$

## A.2    Post-processing inequality under deterministic mappings

**Proposition A.2.** *Let $X, Y$ be random variables taking values in some space $\mathcal{B}$ and let $f : \mathcal{B} \to \mathcal{B}'$ be a measurable function. Then $D_{\infty,\gamma}(f(X), f(Y)) \leq D_{\infty,\gamma}(X,Y)$.*

*Proof.* Fix an arbitrary set $B \subseteq \mathcal{B}'$. We have

$$\log_\gamma \mathbb{P}(f(X) \in B)) - \log_\gamma \mathbb{P}(f(Y) \in B)) = \log_\gamma \mathbb{P}(X \in f^{-1}(B)) - \log_\gamma \mathbb{P}(Y \in f^{-1}(B))$$
$$\leq D_{\infty,\gamma}(X,Y). \qquad\square$$

# B    Proofs for Section 3

This section contains full proofs that are either skipped or simplified in Section 3.

## B.1    Key Lemma

We first record as a lemma the following characterization of $\delta$-approximate max-divergence provided by Dwork et al. [16].

**Lemma B.1.** *[16, Lemma 2.1.1] Let $Y, Z$ be random variables over $\mathcal{B}$. Then, $D_\infty^\delta(Y,Z) \leq \epsilon$, if and only if there exits a random variable $Y'$ such that*

*(i)* $\sup_{B \subseteq \mathcal{B}} |\mathbb{P}[Y \in B] - \mathbb{P}[Y' \in B]| \leq \delta$ *and*
*(ii)* $D_\infty(Y', Z) \leq \epsilon$.

In short, we can alter $Y$ into $Y'$ by moving no more than $\delta$ probability mass from $\{b \in \mathcal{B} : \mathbb{P}[Y = b] > e^\epsilon \mathbb{P}[Z = b]\}$ to $\{b \in \mathcal{B} : \mathbb{P}[Y = b] \leq e^\epsilon \mathbb{P}[Z = b]\}$ such that $D_\infty(Y', Z)$ is bounded. Then in the following lemma, we can show that closeness in max-divergence means that expectations of bounded functions are close. In the result below, when $\delta = 0$, we are allowed to have $F = \infty$.

**Lemma B.2.** *Let $Y$ and $Z$ be random variables taking values in $\mathcal{B}$ such that $D_\infty^\delta(Y, Z) \leq \epsilon$. Then for any non-negative function $f : \mathcal{B} \to [0, F]$, we have*

$$\mathbb{E}[f(Y)] \leq e^\epsilon \mathbb{E}[f(Z)] + \delta F.$$

*Proof.* Let $Y'$ be the random variable satisfying the conditions of Lemma B.1. Then we can write

$$\mathbb{E}[f(Y)] = \int_{\mathcal{B}} f(b)\mathbb{P}[Y = b]db$$

$$= \int_{\mathcal{B}} f(b)\mathbb{P}[Y' = b]db + \int_{\mathcal{B}} f(b)(\mathbb{P}[Y = b] - \mathbb{P}[Y' = b])db$$

$$\leq \int_{\mathcal{B}} f(b)e^{\epsilon}\mathbb{P}[Z = b]db + F|\mathbb{P}[Y \in B] - \mathbb{P}[Y' \in B]|$$

$$\leq e^{\epsilon}\mathbb{E}[f(Z)] + \delta F,$$

where $B = \{b \in \mathcal{B} \mid \mathbb{P}[Y = b] \geq \mathbb{P}[Y' = b])\}$. Here we applied Lemma B.1.(ii) for the first inequality and (i) for the second. □

Now we are ready to prove our key lemma.

**Lemma 3.1.** *Consider the loss-only setting with loss functions bounded by $B$. Let $\mathcal{A}$ be DiffStable($D_{\infty}^{\delta}$) at level $\epsilon \leq 1$. Then the expected regret of $\mathcal{A}$ is at most*

$$2\epsilon L_T^* + 3\mathbb{E}[\mathsf{Regret}(\mathcal{A}^+)_T] + \delta BT,$$

*where $\mathcal{A}^+$ is a fictitious algorithm plays the distribution $\mathcal{A}(\ell_{1:t})$ at time $t$ (i.e., $\mathcal{A}^+$ plays at time $t$ what $\mathcal{A}$ would play at time $t + 1$).*

*Proof.* Let $x_t$ denote the random variable distributed as $\mathcal{A}(\ell_{1:t-1})$. Using Lemma B.2, we have for every $t$, $\mathbb{E}[\ell_t(x_t)] \leq e^{\epsilon}\mathbb{E}[\ell_t(x_{t+1})] + \delta B$. By summing over $t$, we have

$$\mathbb{E}\left[\sum_{t=1}^T \ell_t(x_t)\right] \leq e^{\epsilon}\mathbb{E}\left[\sum_{t=1}^T \ell_t(x_{t+1})\right] + \delta BT \leq e^{\epsilon}(L_T^* + \mathbb{E}[\mathsf{Regret}(\mathcal{A}^+)_T]) + \delta BT.$$

To bound the expected regret of $\mathcal{A}$, we subtract $L_T^*$ from each side, which gives us the bound

$$(e^{\epsilon} - 1)L_T^* + e^{\epsilon}\mathbb{E}[\mathsf{Regret}(\mathcal{A}^+)_T] + \delta BT.$$

Then we complete the proof using the upper bounds $e^{\epsilon} \leq 1 + 2\epsilon \leq 3$, which hold for $\epsilon \leq 1$. □

### B.2  Online convex optimization

**Theorem 3.2** (First-order regret in OCO). *Suppose we are in the loss-only OCO setting. Let $\mathcal{X} \subset \mathbb{R}^d$, $\|\mathcal{X}\|_2 \leq D$ and let all loss functions be bounded by $B$. Further assume that $\|\nabla \ell_t(x)\|_2 \leq \beta$, $\lambda_{\max}(\nabla^2 \ell_t(x)) \leq \gamma$ and that the Hessian matrix $\nabla^2 \ell_t(x)$ has rank at most one, for every $t$ and $x \in \mathcal{X}$. Then, the expected regret of Algorithm 1 is at most $O(\sqrt{L_T^*(\gamma D^2 + \beta dD)})$ and $O\left(\sqrt{L_T^*(\gamma D^2 + D\sqrt{d(\beta^2 \log(BT))})}\right)$ with Gamma and Gaussian perturbations, respectively.*

*Proof.* When analyzing the expected regret against an oblivious adversary, we may assume that the random vector $b$ is just drawn once and reused every round. By definition of $x_t$ and induction on $t$, we get for any $x \in \mathcal{X}$,

$$\frac{\gamma}{\epsilon}\|x_2\|_2^2 + \langle b, x_2 \rangle + \sum_{s=1}^t \ell_s(x_{s+1}) \leq \frac{\gamma}{\epsilon}\|x\|_2^2 + \langle b, x \rangle + \sum_{s=1}^t \ell_s(x).$$

From the case when $t = T$, we obtain

$$\sum_{t=1}^T \ell_t(x_{t+1}) \leq \min_{x \in \mathcal{X}} \sum_{t=1}^T \ell_t(x) + \frac{\gamma}{\epsilon}\|x\|_2^2 + \langle b, x - x_2 \rangle \leq L_T^* + \frac{\gamma}{\epsilon}\|x_T^*\|_2^2 + \langle b, x_T^* - x_2 \rangle.$$

This result is often referred to as the "be-the-leader lemma." Then by taking expectation and applying the Cauchy-Schwartz inequality, we get

$$\mathbb{E}\left[\sum_{t=1}^T \ell_t(x_{t+1})\right] \leq L_T^* + \frac{\gamma}{\epsilon}D^2 + 2D\mathbb{E}\|b\|_2.$$

From the DP result by Kifer et al. [23, Theorem 2], we can infer that $D_\infty(x_t, x_{t+1}) \leq \epsilon$ when using the Gamma distribution and $D_\infty^\delta(x_t, x_{t+1}) \leq \epsilon$ when using the Gaussian distribution. This means that Algorithm 1 enjoys the one-step differential stability w.r.t. $D_\infty$ (resp. $D_\infty^\delta$) in the Gamma (resp. Gaussian) case. Using Lemma 3.1, we can deduce that the expected regret of Algorithm 1 is at most

$$2\epsilon L_T^* + \frac{3\gamma}{\epsilon} D^2 + 6D\mathbb{E}\|b\|_2 + \delta BT, \tag{5}$$

where $\delta$ becomes zero when using the Gamma distribution. We have $\|b\|_2 \sim \text{Gamma}(d, \frac{\epsilon}{2\beta})$ when using the Gamma distribution, which gives $\mathbb{E}\|b\|_2 = \frac{2\beta d}{\epsilon}$. In the case of the Gaussian distribution, we have $\mathbb{E}\|b\|_2 \leq \sqrt{\mathbb{E}\|b\|_2^2} = \frac{\sqrt{d(\beta^2 \log \frac{2}{\delta} + 4\epsilon)}}{\epsilon}$. Plugging these results in (5) and optimizing $\epsilon$ (setting $\delta = \frac{1}{BT}$ for the Gaussian case) prove the desired bound. $\square$

### B.3 Differential consistency and one-step differential stability

We first prove the claim that we made in Section 3.3.1.

**Proposition B.3.** *Suppose $\tilde{\Phi}(L)$ is of the form $\min_p \langle L, p \rangle + F(p)$ for a separable $F(p) = \sum_{i=1}^N f(p_i)$ with $f : (0, \infty) \to \mathbb{R}$ differentiable and strictly convex. Then, the matrix $-\nabla^2 \tilde{\Phi}(L)$ is POD for any L.*

*Proof.* We have the gradient formula

$$g(L) = \nabla \tilde{\Phi}(L) = \underset{p \in \Delta^{N-1}}{\arg\min} \sum_{i=1}^N p_i L_i + \sum_{i=1}^N f(p_i).$$

Let $\lambda = \lambda(L)$ be the Lagrange multiplier for the constraint $\sum_i p_i = 1$. We do not have to worry about the constraint $p_i \geq 0$ since we have assumed that the domain of $f$ is $(0, \infty)$. Setting the derivative of the Lagrangian

$$\sum_i p_i L_i + \sum_i f(p_i) + \lambda(\sum_i p_i - 1)$$

w.r.t. $p_i$ to zero gives us

$$L_i + f'(p_i) + \lambda = 0. \tag{6}$$

Taking the derivatives w.r.t. $L_i$ and $L_j$ for $j \neq i$ gives us

$$f''(p_i) \frac{\partial p_i}{\partial L_i} = -1 - \frac{\partial \lambda}{\partial L_i}$$

$$f''(p_i) \frac{\partial p_i}{\partial L_j} = -\frac{\partial \lambda}{\partial L_j}.$$

Now note that $f'' > 0$ ($f$ strictly convex) and $\nabla_{ij}^2 \tilde{\Phi}(L) = \frac{\partial p_i}{\partial L_j}$. The proof will therefore be complete if we can claim that $-\frac{\partial \lambda}{\partial L_i} \in (0, 1)$. Let us next prove this claim. We can write (6) as

$$p_i = (f^\star)'(-\lambda - L_i),$$

where $f^\star$ is the Fenchel conjugate of $f$ (and therefore $f'$ and $(f^\star)'$ are inverses of each other). Plugging this into the constraint $\sum_i p_i = 1$ gives

$$\sum_i (f^\star)'(-\lambda - L_i) = 1.$$

Now differentiating w.r.t. $L_i$ gives us

$$(f^\star)''(-\lambda - L_i)\left(-\frac{\partial \lambda}{\partial L_i} - 1\right) + \sum_{j \neq i}(f^\star)''(-\lambda - L_j)\left(-\frac{\partial \lambda}{\partial L_i}\right) = 0,$$

which upon rearranging yields

$$-\frac{\partial \lambda}{\partial L_i} = \frac{(f^\star)''(-\lambda - L_i)}{\sum_j (f^\star)''(-\lambda - L_j)}.$$

Since $f$ is smooth, $f^\star$ is strictly convex and therefore $(f^\star)'' > 0$ which proves $-\frac{\partial \lambda}{\partial L_i} \in (0, 1)$. $\square$

Table 1: The parameter settings for the distributions that provide $(1, \epsilon)$-differential consistency of the induced potentials while keeping $\mathbb{E}_{Z_1, \cdots Z_N \sim \mathcal{D}} \max_i Z_i = O(\log N / \epsilon)$ [2]. Distributions marked with a $*$ have to be modified slightly to ensure the differential consistency.

| Distribution $\mathcal{D}$ | Parameter choice |
|---|---|
| Gamma$(\alpha, \beta)$ | $\alpha = 1, \beta = 1$ |
| Gumbel$(\mu, \beta)$ | $\mu = 0, \beta = 1$ |
| Fréchet $(\alpha > 1)$ | $\alpha = \log N$ |
| Weibull$^*(\lambda, k)$ | $\lambda = 1, k = 1$ |
| Pareto$^*(x_m, \alpha)$ | $x_m = 1, \alpha = \log N$ |

Next we prove Proposition 3.5.

**Proposition 3.5** (Differential consistency implies one-step differential stability). *Suppose $\tilde{\Phi}(L)$ is of the form $\mathbb{E}[\min_p \langle a, p \rangle + F(p, Z)]$ and $\gamma \geq 1$. If $\tilde{\Phi}$ is $(\gamma, \epsilon)$-differentially consistent and $-\nabla^2 \tilde{\Phi}$ is always POD, the GBPA using $\tilde{\Phi}$ as potential is DiffStable($D_{\infty, \gamma}$, $\| \cdot \|_\infty$) at level $2\epsilon$.*

*Proof.* First, note that by the POD property, the second derivative vector $\nabla^2_{i\cdot} \tilde{\Phi} = (\nabla^2_{i1} \tilde{\Phi}, \ldots, \nabla^2_{iN} \tilde{\Phi})$ satisfies that the $i$-th coordinate is non-positive and the rest are non-negative. Next, because the entries in the gradient sum to a constant (it is a probability vector), we know that the coordinate of the second derivative vector add up to 0 [1]. From this, we can write $\|\nabla^2_{i\cdot} \tilde{\Phi}\|_1 = -2\nabla^2_{ii} \tilde{\Phi}$. Let the cumulative sum of losses so far be $L$ and the new loss vector be $\ell$. For $P = \nabla \tilde{\Phi}(L), Q = \nabla \tilde{\Phi}(L + \ell)$, we want to show that $D_{\infty, \gamma}(P, Q) \leq 2\epsilon \|\ell\|_\infty$. To this end, fix a subset $S \subseteq [N]$ and define $q_S(u) = \sum_{i \in S} \nabla_i \tilde{\Phi}(L + \ell - u\ell)$ for $u \in [0, 1]$. Its derivative can be written as

$$q'_S(u) = \sum_{i \in S} \langle \nabla^2_{i\cdot} \tilde{\Phi}(L + \ell - u\ell), -\ell \rangle$$

$$\leq \sum_{i \in S} \|\nabla^2_{i\cdot} \tilde{\Phi}(L + \ell - u\ell)\|_1 \|\ell\|_\infty = \sum_{i \in S} -2\nabla^2_{ii} \tilde{\Phi}(L + \ell - u\ell) \|\ell\|_\infty$$

$$\leq 2\epsilon \|\ell\|_\infty \sum_{i \in S} \left( \nabla_i \tilde{\Phi}(L + \ell - u\ell) \right)^\gamma$$

$$\leq 2\epsilon \|\ell\|_\infty \left( \sum_{i \in S} \nabla_i \tilde{\Phi}(L + \ell - u\ell) \right)^\gamma = 2\epsilon \|\ell\|_\infty (q_S(u))^\gamma.$$

The first inequality follows from duality of $\ell_1, \ell_\infty$ norms. The second inequality is from our differential consistency assumption. The last inequality holds because gradient has non-negative entries and $\| \cdot \|_\gamma \leq \| \cdot \|_1$ for $\gamma \geq 1$. It follows that for any $u \in [0, 1]$, we have

$$\frac{q'_S(u)}{(q_S(u))^\gamma} = \frac{d}{du} \log_\gamma (q_S(u)) \leq 2\epsilon \|\ell\|_\infty,$$

and therefore

$$\log_\gamma(P(S)) - \log_\gamma(Q(S)) = \log_\gamma q_S(1) - \log_\gamma q_S(0) = \int_0^1 \frac{d}{du} \log_\gamma (q_i(u)) \, du$$

$$\leq 2\epsilon \|\ell\|_\infty. \qquad \square$$

Now we are ready to prove the first order bound of FTPL in the experts problem.

**Theorem 3.6** (First-order bound for experts via FTPL). *For the loss-only experts setting, FTPL with Gamma, Gumbel, Fréchet , Weibull, and Pareto perturbations, with a proper choice of distribution parameters, all achieve the optimal $O(\sqrt{L_T^* \log N} + \log N)$ expected regret.*

*Proof.* Recall that the FTPL algorithm uses the potential $\tilde{\Phi}_{\mathcal{D}}(L) = \mathbb{E}_{Z_1, \cdots, Z_N \sim \mathcal{D}} \min_i (L_i - Z_i)$. Abernethy et al. [2] show that all the listed distributions, after suitable scaling, have this potential $\tilde{\Phi}_{\mathcal{D}}$ $(1, \epsilon)$-differentially consistent and $\mathbb{E}\|Z\|_\infty = O(\log N / \epsilon)$ at the same time (for parameter choices,

---

**Algorithm 4** GBPA for multi-armed bandits problem

1: **Input:** Concave potential $\tilde{\Phi} : \mathbb{R}^N \to \mathbb{R}$, with $\nabla \tilde{\Phi} \in \Delta^{N-1}$
2: Set $\hat{L}_0 = 0 \in \mathbb{R}^N$
3: **for** $t = 1$ to $T$ **do**
4:     **Sampling:** Choose $i_t \in [N]$ according to distribution $p_t = \nabla \tilde{\Phi}(\hat{L}_{t-1}) \in \Delta^{N-1}$
5:     **Loss:** Incur loss $\ell_{t,i_t}$ and observe this value
6:     **Estimation:** Compute an estimate of loss vector, $\hat{\ell}_t = \frac{\ell_{t,i_t}}{p_{t,i_t}} \mathbf{e}_{i_t}$
7:     **Update:** $\hat{L}_t = \hat{L}_{t-1} + \hat{\ell}_t$
8: **end for**

---

see Table 1). Then Proposition 3.5 provides that FTPL with any of the listed distributions is one-step differentially stable with respect to $\|\cdot\|_\infty$ and $D_\infty$ at level $2\epsilon$. Using the "be-the-leader lemma" (as in the proof of Theorem 3.2), we have

$$\mathbb{E}[\sum_{t=1}^T \ell_{t,i_{t+1}}] - L_T^* \leq 2\mathbb{E}\|Z\|_\infty = O(\log N/\epsilon).$$

Applying Lemma 3.1 with $\epsilon = \min(\sqrt{\log N/L_T^*}, 1)$ completes the proof. $\qquad\square$

## C  Details and proofs for Section 4

Here we present missing parts in Section 4. We start by presenting the GBPA for multi-armed bandits in Algorithm 4.

### C.1  Proof of Lemma 4.1

We will prove the following slightly more general lemma. Lemma 4.1 follows by setting $\tau = 0$ and lower bounding $F(p_0, Z)$ by $\min_p F(p, Z)$.

**Lemma C.1.** *Suppose the full information GBPA uses a potential of the form $\tilde{\Phi}(L) = \mathbb{E}[\min_p \langle L, p \rangle + F(p, Z)]$ and $\gamma \in [1, 2]$. If the full information GBPA is DiffStable($D_{\infty,\gamma}, \|\cdot\|_\infty$) at level $\epsilon$, then the expected regret of Algorithm 4 can be bounded as*

$$\mathbb{E}\left[\sum_{t=1}^T \ell_{t,i_t}\right] - L_T^* \leq \epsilon\, \mathbb{E}\left[\sum_{t=1}^T \hat{\ell}_{t,i_t}^2 p_{t,i_t}^\gamma\right] + \mathbb{E}\left[\max_{p \in \Delta_\tau} F(p, Z) - F(p_0, Z)\right] + \tau N T,$$

*where $\Delta_\tau = \{p \in \mathbb{R}^N : p_i \geq \tau, \sum_i p_i = 1\}$.*

We will prove this lemma by proving two helper lemmas (Lemma C.2 and Lemma C.3 below) that, when combined, immediately yield the desired result.

**Lemma C.2.** *Suppose the full information GBPA uses a potential of the form $\tilde{\Phi}(L) = \mathbb{E}[\min_p \langle L, p \rangle + F(p, Z)]$. Then, we have*

$$\mathbb{E}\left[\sum_{t=1}^T \ell_{t,i_t}\right] - L_T^* \leq \mathbb{E}\left[\sum_{t=1}^T \langle p_t - p_{t+1}, \hat{\ell}_t \rangle\right] + \mathbb{E}\left[\max_{p \in \Delta_\tau} F(p, Z) - F(p_0, Z)\right] + \tau N T$$

*where $\Delta_\tau = \{p \in \mathbb{R}^N : p_i \geq \tau, \sum_i p_i = 1\}$.*

*Proof.* Fix the source of internal randomness used by Algorithm 4 to sample $i_t \sim p_t$. This fixes all the estimated loss vectors $\hat{\ell}_t$. The full information GBPA algorithm will deterministically generate the same sequence $p_t$ of probabilities on this estimated loss sequence using the rule $p_t = \nabla\tilde{\Phi}(\hat{L}_{t-1})$. We have

$$\sum_{t=1}^T \langle p_t, \hat{\ell}_t \rangle = \sum_{t=1}^T \langle p_t - p_{t+1}, \hat{\ell}_t \rangle + \sum_{t=1}^T \langle p_{t+1}, \hat{\ell}_t \rangle. \tag{7}$$

Let us focus on the second summation:

$$\sum_{t=1}^{T} \langle p_{t+1}, \hat{\ell}_t \rangle = \sum_{t=1}^{T} \langle \mathbb{E}[\arg\min_{p} \langle \hat{L}_t, p \rangle + F(p, Z)], \hat{\ell}_t \rangle$$

$$= \mathbb{E}[\sum_{t=1}^{T} \langle \arg\min_{p} \langle \hat{L}_t, p \rangle + F(p, Z), \hat{\ell}_t \rangle]$$

$$\leq \mathbb{E}[F(p, Z) - F(p_0, Z)] + \sum_{t=1}^{T} \langle p, \hat{\ell}_t \rangle, \qquad (8)$$

where the last inequality is true for any $p$ due to the "be-the-leader" argument. Note that the expectations above are only w.r.t. $Z$ since $\hat{\ell}_t$ are still fixed. Combining (7) and (8) and taking expectations over the internal randomness of the bandit algorithm gives, for any $p \in \Delta_\tau$,

$$\mathbb{E}[\sum_{t=1}^{T} \langle p_t, \hat{\ell}_t \rangle] \leq \mathbb{E}[\sum_{t=1}^{T} \langle p_t - p_{t+1}, \hat{\ell}_t \rangle] + \mathbb{E}[F(p, Z) - F(p_0, Z)] + \mathbb{E}[\sum_{t=1}^{T} \langle p, \hat{\ell}_t \rangle]$$

$$= \mathbb{E}[\sum_{t=1}^{T} \langle p_t - p_{t+1}, \hat{\ell}_t \rangle] + \mathbb{E}[F(p, Z) - F(p_0, Z)] + \sum_{t=1}^{T} \langle p, \ell_t \rangle$$

$$\leq \mathbb{E}[\sum_{t=1}^{T} \langle p_t - p_{t+1}, \hat{\ell}_t \rangle] + \mathbb{E}[\max_{p \in \Delta_\tau} F(p, Z) - F(p_0, Z)] + \sum_{t=1}^{T} \langle p, \ell_t \rangle.$$

To finish the proof, note that $\langle p_t, \hat{\ell}_t \rangle = \ell_{t,i_t}$ and

$$\min_{p \in \Delta_\tau} \sum_{t=1}^{T} \langle p, \ell_t \rangle \leq \min_{p} \sum_{t=1}^{T} \langle p, \ell_t \rangle + \tau N T = L_T^* + \tau N T.$$

because for any $p' \in \Delta^{N-1}$, there is a $p \in \Delta_\tau$ such that $\|p' - p\|_1 \leq \tau N$. $\qquad \square$

**Lemma C.3.** *Suppose $1 \leq \gamma \leq 2$. If the full information GBPA is DiffStable($D_{\infty,\gamma}, \| \cdot \|_\infty$) at level $\epsilon$. Then, we have*

$$\langle p_t - p_{t+1}, \hat{\ell}_t \rangle \leq \epsilon \hat{\ell}_{t,i_t}^2 p_{t,i_t}^\gamma.$$

*Proof.* First let us consider the case $\gamma > 1$ first. Recall at most one entry of $\hat{\ell}_t$ is non-zero. Therefore, $\langle p_t - p_{t+1}, \hat{\ell}_t \rangle = (p_{t,i_t} - p_{t+1,i_t}) \hat{\ell}_{t,i_t}$. For the remainder of the proof, let us denote $p_{t,i_t}$ and $p_{t+1,i_t}$ by $p$ and $p'$ repectively. Also let $\hat{\ell}$ denote $\hat{\ell}_{t,i_t}$. Because of the stability assumption, we know that $D_{\infty,\gamma}(p_t, p_{t+1}) \leq \epsilon \|\hat{\ell}_t\|_\infty = \epsilon \hat{\ell}$. Therefore, we have

$$\log_\gamma p - \log_\gamma p' \leq \epsilon \hat{\ell}$$

$$\Rightarrow \qquad \frac{p^{1-\gamma}}{1-\gamma} - \frac{(p')^{1-\gamma}}{1-\gamma} \leq \epsilon \hat{\ell}$$

$$\Rightarrow \qquad p' \geq p \left(1 + (\gamma-1)\epsilon \hat{\ell} p^{\gamma-1}\right)^{-\frac{1}{\gamma-1}}.$$

Noting that $(1+x)^{-r} \geq 1 - rx$ for $r > 0$ and $x \geq 0$, we have

$$p' \geq p \left(1 - \epsilon \hat{\ell} p^{\gamma-1}\right),$$

which proves the lemma for $1 < \gamma \leq 2$.

Finally note that the lemma also holds in the case $\gamma = 1$, because then we have

$$\log p - \log p' \leq \epsilon \hat{\ell}$$

$$\Rightarrow \qquad p' \geq p \exp(-\epsilon \hat{\ell}) \geq p(1 - \epsilon \hat{\ell}). \qquad \square$$

## C.2 Proof of Theorem 4.2

**Theorem 4.2** (Zero-order and first-order regret bounds for multi-armed bandits). *Algorithm 4 enjoys the following bounds when used with different perturbations/regularizers:*

1. *FTPL with Gamma, Gumbel, Fréchet , Weibull, and Pareto pertubations (with a proper choice of distribution parameters) all achieve near-optimal expected regret of $O(\sqrt{NT \log N})$.*
2. *FTRL with Tsallis neg-entropy $F(p) = -\eta \sum_{i=1}^{N} p_i \log_\alpha(1/p_i)$ for $0 < \alpha < 1$ (with a proper choice of $\eta$) achieves optimal expected regret of $O(\sqrt{NT})$.*
3. *FTRL with log-barrier regularizer $F(p) = -\eta \sum_{i=1}^{N} \log p_i$ (with a proper choice of $\eta$) achieves expected regret of $O(\sqrt{NL_T^* \log(NT)} + N \log(NT))$.*

*Proof.* For each of the three parts, we will show that the full info GBPA is DiffStable($D_{\infty,\gamma}, \|\cdot\|_\infty$) for an appropriate $\gamma \in [1, 2]$ and then apply Lemma 4.1 (or, in the log-barrier case, the slightly more general Lemma C.1).

**Part 1:** As in the proof of Theorem 3.6, these distributions (with proper choice of parameters) lead to a full information GBPA that is DiffStable($D_\infty, \|\cdot\|_\infty$) at level $2\epsilon$. In the FTPL case when $|F(p, Z)| = |\langle p, Z \rangle| \leq \|Z\|_\infty$, we have

$$\mathbb{E}\left[\max_p F(p, Z) - \min_p F(p, Z)\right] \leq 2\mathbb{E}\|Z\|_\infty,$$

which scales as $\frac{\log N}{\epsilon}$ for these distributions. Lemma 4.1 gives the expected regret bound of

$$2\epsilon \, \mathbb{E}\left[\sum_{t=1}^{T} \hat{\ell}_{t,i_t}^2 p_{t,i_t}\right] + 2\mathbb{E}\|Z\|_\infty. \tag{9}$$

Since $\ell_{t,i_t} \in [0, 1]$, we have

$$\mathbb{E}\left[\hat{\ell}_{t,i_t}^2 p_{t,i_t}\right] = \mathbb{E}\left[\frac{\ell_{t,i_t}^2}{p_{t,i_t}^2} p_{t,i_t}\right] \leq \mathbb{E}\left[\frac{1}{p_{t,i_t}}\right] = \mathbb{E}\left[\sum_{i=1}^{N} p_{t,i} \frac{1}{p_{t,i}}\right] = N.$$

Plugging this into (9) and tuning $\epsilon$ give us Part 1.

**Part 2:** When $F(p) = -\eta \sum_{i=1}^{N} p_i \log_\alpha(1/p_i)$ for $\alpha \in (0,1)$, then $\tilde{\Phi}$ is $(2 - \alpha, 1/(\eta\alpha))$-differentially consistent [2]. By Proposition 3.5, the full information GBPA is DiffStable($D_{\infty,2-\alpha}, \|\cdot\|_\infty$) at level $2/(\eta\alpha)$. Also note that $F(p)$ is negative and its minimum value is achieved at the uniform distribution. Therefore we can show

$$\max_p F(p) - \min_p F(p) \leq \eta \frac{\sum_{i=1}^{N}(1/N)^\alpha - 1}{1 - \alpha} \leq \eta \frac{N^{1-\alpha}}{1 - \alpha}.$$

Lemma 4.1 gives the expected regret bound of

$$\frac{2}{\eta\alpha} \, \mathbb{E}\left[\sum_{t=1}^{T} \hat{\ell}_{t,i_t}^2 p_{t,i_t}^{2-\alpha}\right] + \eta \frac{N^{1-\alpha}}{1 - \alpha}. \tag{10}$$

Since $\ell_{t,i_t} \in [0, 1]$, we have

$$\mathbb{E}\left[\hat{\ell}_{t,i_t}^2 p_{t,i_t}^{2-\alpha}\right] = \mathbb{E}\left[\frac{\ell_{t,i_t}^2}{p_{t,i_t}^2} p_{t,i_t}^{2-\alpha}\right] \leq \mathbb{E}\left[p_{t,i_t}^{-\alpha}\right] = \mathbb{E}\left[\sum_{i=1}^{N} p_{t,i}^{1-\alpha}\right] \leq N^\alpha,$$

where the last inequality follows from the fact that the function $p^{1-\alpha}$ is concave for $\alpha \in (0,1)$. Plugging this into (10) and tuning $\eta$ give us Part 2.

**Part 3:** When $F(p) = -\eta \sum_{i=1}^{N} \log p_i$, then $\tilde{\Phi}$ is $(2, 1/\eta)$-differentially consistent (see Lemma C.4 below). By Proposition 3.5, the full information GBPA is DiffStable($D_{\infty,2}, \|\cdot\|_\infty$) at level $2/\eta$. Since $F$ is non-negative, we have

$$\max_{p \in \Delta_\tau} F(p) - F(p_0) \leq \eta N \log(1/\tau).$$

Lemma C.1 gives the expected regret bound of

$$\frac{2}{\eta} \mathbb{E}\left[\sum_{t=1}^{T} \hat{\ell}_{t,i_t}^2 p_{t,i_t}^2\right] + \eta N \log(1/\tau) + \tau NT. \tag{11}$$

Since $\ell_{t,i_t} \in [0,1]$, we have

$$\mathbb{E}\left[\hat{\ell}_{t,i_t}^2 p_{t,i_t}^2\right] = \mathbb{E}\left[\frac{\ell_{t,i_t}^2}{p_{t,i_t}^2} p_{t,i_t}^2\right] \leq \mathbb{E}\left[\ell_{t,i_t}^2\right] \leq \mathbb{E}\left[\ell_{t,i_t}\right].$$

Plugging this into (11) and choosing $\tau = 1/(NT)$, we get the following recursive inequality for the expected regret $R_T$:

$$R_T \leq \frac{2}{\eta}\left(R_T + L_T^*\right) + \eta N \log(NT) + 1.$$

If $L_T^* < 2$, set $\eta = 4$ to get the bound $R_T \leq 8N \log(NT) + 4$. If $L_T^* \geq 2$ then set $\eta = \sqrt{2L_T^* N \log(NT)}$ and note that $\eta > 2\sqrt{2}$ if $N, T \geq 2$. In this case, the bound becomes $R_T \leq \left(4\sqrt{L_T^* N \log(NT)} + 1\right)/(\sqrt{2} - 1)$, which completes the proof. $\qquad\square$

## C.3 Differential consistency of GBPA potential with log barrier regularization

**Lemma C.4.** *Let* $F(p) = -\eta \sum_{i=1}^{N} \log p_i$, $\tilde{\Phi}(L) = \min_p \langle L, p \rangle + F(p)$ *and* $p(L) = \arg\min_p \langle L, p \rangle + F(p)$. *Then* $\tilde{\Phi}$ *is* $(2, 1/\eta)$-*differentially consistent.*

*Proof.* We observe that straightforward calculus gives $\nabla^2 F(p) = \eta \operatorname{diag}(p_1^{-2}, \ldots, p_N^{-2})$. Let $\mathbb{I}_{\Delta^{N-1}}(\cdot)$ be the indicator function of $\Delta^{N-1}$; that is, $\mathbb{I}_{\Delta^{N-1}}(x) = 0$ for $x \in \Delta^{N-1}$ and $\mathbb{I}_{\Delta^{N-1}}(x) = \infty$ for $x \notin \Delta^{N-1}$. It is clear that $-\tilde{\Phi}(-L)$ is the dual of the function $F(x) + \mathbb{I}_{\Delta^{N-1}}(x)$, and moreover we observe that $\nabla^2 F(p)$ is a *sub-Hessian* of $F(\cdot) + \mathbb{I}_{\Delta_N}(\cdot)$ at $p$, following the setup of [30]. Taking advantage of Proposition 3.2 in the latter reference, we conclude that $\nabla^{-2} F(p(-L))$ is a *super-Hessian* of $F^* = -\tilde{\Phi}(-L)$ at $L$. Hence, we get

$$-\nabla^2 \tilde{\Phi}(-L) \preceq \eta^{-1} \operatorname{diag}(p_1^2(-L), \ldots, p_N^2(-L))$$

for any $L$. What we have stated, indeed, is that $\tilde{\Phi}$ is $(2, 1/\eta)$-differentially-consistent. $\qquad\square$

## C.4 Bandits with experts

We provide full details for the bandits with experts setting.

### C.4.1 Helper lemmas for the analysis

**Lemma C.5.** *Suppose the full information GBPA uses a potential of the form* $\tilde{\Phi}(L) = \mathbb{E}[\min_p \langle L, p \rangle + F(p, Z)]$. *If the full information GBPA is DiffStable($D_\infty, \|\cdot\|_\infty$) at level $\epsilon$, then the expected regret of Algorithm 3 with no clipping can be bounded as*

$$\mathbb{E}\left[\sum_{t=1}^{T} \ell_{t,j_t}\right] - L_T^* \leq \epsilon \mathbb{E}\left[\sum_{t=1}^{T} \hat{\ell}_{t,i_t}^2 q_{t,j_t}\right] + \mathbb{E}\left[\max_p F(p, Z) - \min_p F(p, Z)\right].$$

We will prove this lemma by proving Lemma C.6 and Lemma C.7 below that, when combined, immediately yield the desired result.

**Lemma C.6.** *Suppose the full information GBPA uses a potential of the form* $\tilde{\Phi}(L) = \mathbb{E}[\min_p \langle L, p \rangle + F(p, Z)]$. *Then, we have*

$$\mathbb{E}\left[\sum_{t=1}^{T} \ell_{t,i_t}\right] - L_T^* \leq \mathbb{E}\left[\sum_{t=1}^{T} \langle p_t - p_{t+1}, \overline{\phi}_t \rangle\right] + \mathbb{E}\left[\max_p F(p, Z) - \min_p F(p, Z)\right].$$

*Proof.* Fix the source of internal randomness used by Algorithm 3 (with no clipping) to sample $j_t \sim q_t$. This fixes all the estimated loss vectors $\hat{\ell}_t$ and hence $\overline{\phi}_t$. The full information GBPA algorithm will deterministically generate the same sequence $p_t$ of probabilities on this estimated loss sequence using the rule $p_t = \nabla\tilde{\Phi}(\overline{\phi}_{t-1})$. We have

$$\sum_{t=1}^{T}\langle p_t, \overline{\phi}_t\rangle = \sum_{t=1}^{T}\langle p_t - p_{t+1}, \overline{\phi}_t\rangle + \sum_{t=1}^{T}\langle p_{t+1}, \overline{\phi}_t\rangle.$$

Proceeding as in the proof of Lemma C.2 gives us, for any $p \in \Delta^{N-1}$,

$$\mathbb{E}[\sum_{t=1}^{T}\langle p_t, \overline{\phi}_t\rangle] \le \mathbb{E}[\sum_{t=1}^{T}\langle p_t - p_{t+1}, \overline{\phi}_t\rangle] + \mathbb{E}[\max_p F(p, Z) - \min_p F(p, Z)] + \mathbb{E}[\sum_{t=1}^{T}\langle p, \overline{\phi}_t\rangle].$$

To finish the proof, first note that

$$\mathbb{E}[\sum_{t=1}^{T}\langle p_t, \overline{\phi}_t\rangle] = \mathbb{E}\left[\sum_{t=1}^{T}\langle p_t, \phi_t(\hat{\ell}_t)\rangle\right] = \mathbb{E}\left[\sum_{t=1}^{T}\langle \psi_t(p_t), \hat{\ell}_t\rangle\right]$$
$$= \mathbb{E}\left[\sum_{t=1}^{T}\langle q_t, \hat{\ell}_t\rangle\right] = \mathbb{E}\left[\sum_{t=1}^{T}\ell_{t,j_t}\right].$$

Second, note that, by choosing $p = \mathbf{e}_i \in \mathbb{R}^N$,

$$\sum_{t=1}^{T}\mathbb{E}[\langle p, \overline{\phi}_t\rangle] = \sum_{t=1}^{T}\mathbb{E}[\langle \mathbf{e}_i, \overline{\phi}_t\rangle] = \sum_{t=1}^{T}\mathbb{E}[\langle e_i, \phi_t(\hat{\ell}_t)\rangle]$$
$$= \sum_{t=1}^{T}\mathbb{E}\left[\langle \psi_t(e_i), \hat{\ell}_t\rangle\right] = \sum_{t=1}^{T}\langle \psi_t(e_i), \ell_t\rangle = \sum_{t=1}^{T}\ell_{t,E_{i,t}}.$$

Choosing $i$ that makes the last summation equal to $L_T^*$ completes the proof. $\qquad\square$

**Lemma C.7.** *If the full information GBPA is DiffStable($D_\infty, \|\cdot\|_\infty$) at level $\epsilon$. Then, we have*

$$\langle p_t - p_{t+1}, \overline{\phi}_t\rangle \le \epsilon\hat{\ell}_{t,j_t}^2 q_{t,j_t}.$$

*Proof.* Because of the stability assumption, we know that $D_{\infty,\gamma}(p_t, p_{t+1}) \le \epsilon\|\overline{\phi}_t\|_\infty = \epsilon\|\hat{\ell}_t\|_\infty = \epsilon\hat{\ell}_{t,j_t}$. Therefore, for any $i \in [N]$,

$$p_{t+1,i} \ge p_{t,i}\exp(-\epsilon\hat{\ell}_{t,j_t}) \ge p_{t,i}(1 - \epsilon\hat{\ell}_{t,j_t}).$$

Now we have

$$\langle p_t - p_{t+1}, \phi_t(\hat{\ell}_t)\rangle = \sum_{i:E_{i,t}=j_t}(p_{t,i} - p_{t+1,i})\hat{\ell}_{t,j_t} \le \sum_{i:E_{i,t}=j_t}(\epsilon\hat{\ell}_{t,j_t}p_{t,i})\hat{\ell}_{t,j_t}$$
$$= \epsilon\hat{\ell}_{t,j_t}^2\sum_{i:E_{i,t}=j_t}p_{t,i} = \epsilon\hat{\ell}_{t,j_t}^2 q_{t,j_t}. \qquad\square$$

### C.4.2 Regret bound and analysis

**Theorem 4.3** (Zero-order and first-order regret bounds for bandits with experts). *Algorithm 3 enjoys the following bounds when used with different perturbations such as Gamma, Gumbel, Fréchet , Weibull, and Pareto (with a proper choice of parameters).*

1. *With no clipping, it achieves near optimal expected regret of $O(\sqrt{KT\log N})$.*
2. *With clipping, it achieves expected regret of $O\left((K\log N)^{1/3}(L_T^*)^{2/3}\right)$.*

*Proof.* The proof of Part 1 is very similar to the proof of Part 1 of Theorem 4.2. Part 2 needs a few more arguments to take care of the effects of clipping.

**Part 1:** Note that without clipping, i.e., when $\rho = 0$, we have $q_t = \tilde{q}_t$ for all $t$. As in the proof of Theorem 3.6, these distributions (with proper choice of parameters) lead to a full information GBPA that is DiffStable($D_\infty, \|\cdot\|_\infty$) at level $2\epsilon$. In the FTPL case when $|F(p, Z)| = |\langle p, Z\rangle| \leq \|Z\|_\infty$, we have

$$\mathbb{E}\left[\max_p F(p, Z) - \min_p F(p, Z)\right] \leq 2\mathbb{E}\|Z\|_\infty,$$

which scales as $\frac{\log N}{\epsilon}$ for these distributions. Lemma C.5 gives the expected regret bound of

$$2\epsilon\, \mathbb{E}\left[\sum_{t=1}^T \hat{\ell}_{t,i_t}^2 p_{t,i_t}\right] + 2\mathbb{E}\|Z\|_\infty. \tag{12}$$

Since $\ell_{t,j_t} \in [0, 1]$, we have

$$\mathbb{E}\left[\hat{\ell}_{t,i_t}^2 q_{t,j_t}\right] = \mathbb{E}\left[\frac{\ell_{t,i_t}^2}{q_{t,j_t}^2} q_{t,j_t}\right] \leq \mathbb{E}\left[\frac{1}{q_{t,j_t}}\right] = \epsilon\mathbb{E}\left[\sum_{i=1}^K q_{t,j}\frac{1}{q_{t,j}}\right] = \epsilon K.$$

Plugging this into (12) and tuning $\epsilon$ gives us Part 1.

**Part 2:** When there is clipping, i.e. $\rho$ is non-zero, the loss estimate behaves differently from estimate used in the unclipped version of the algorithm. First, we have the upper bound $\|\phi_t(\hat{\ell}_t)\|_\infty = \|\hat{\ell}_t\|_\infty \leq 1/\rho$. Second, it is unbiased only over the support of $\tilde{q}_t$ since outside of the support, it is deterministically zero. Therefore $\hat{\ell}_t$, in expectation, now *underestimates* $\ell_t$. Crucially, however, we still have the equality $\mathbb{E}[\langle\tilde{q}_t, \hat{\ell}_t\rangle] = \mathbb{E}[\ell_{t,j_t}]$.

Lemma C.6 does not directly bound regret in the clipped case. However, examining the proof, it gives us the bound,

$$\mathbb{E}\left[\sum_{t=1}^T \langle p_t, \phi_t(\hat{\ell}_t)\rangle\right] - \min_{i=1}^N \sum_{t=1}^T \mathbb{E}\left[\langle\psi_t(e_i), \hat{\ell}_t\rangle\right]$$

$$\leq \mathbb{E}\left[\sum_{t=1}^T \langle p_t - p_{t+1}, \overline{\phi}_t\rangle\right] + \mathbb{E}\left[\max_p F(p, Z) - \min_p F(p, Z)\right]. \tag{13}$$

We will now relate the LHS to regret and bound the RHS.

First, note that

$$\mathbb{E}\left[\sum_{t=1}^T \langle p_t, \phi_t(\hat{\ell}_t)\rangle\right] = \mathbb{E}\left[\sum_{t=1}^T \langle q_t, \hat{\ell}_t\rangle\right] \geq (1-K\rho)\mathbb{E}\left[\sum_{t=1}^T \langle\tilde{q}_t, \hat{\ell}_t\rangle\right] = (1-K\rho)\mathbb{E}\left[\sum_{t=1}^T \ell_{t,j_t}\right] \tag{14}$$

where the inequality follows because $\hat{\ell}_t$ has all non-negative entries and $\tilde{q}_{t,j}$ is either 0 or we have

$$\tilde{q}_{t,j} = \frac{q_{t,j}}{1 - \sum_{j':q_{t,j'}<\rho} q_{t,j'}} \leq \frac{q_{t,j}}{1 - K\rho}.$$

Second, note that, because $\hat{\ell}_t$ underestimates $\ell_t$, we have

$$\mathbb{E}\left[\sum_{t=1}^T \phi_{t,i}(\hat{\ell}_t)\right] = \sum_{t=1}^T \mathbb{E}[\langle e_i, \phi_t(\hat{\ell}_t)\rangle] = \sum_{t=1}^T \mathbb{E}\left[\langle\psi_t(e_i), \hat{\ell}_t\rangle\right] \leq \sum_{t=1}^T \langle\psi_t(e_i), \ell_t\rangle = \sum_{t=1}^T \ell_{t,E_{i,t}}. \tag{15}$$

Third, note that, as in the proof of Theorem 3.6, these distributions (with proper choice of parameters) lead to a DiffStable($D_\infty, \|\cdot\|_\infty$) full information GBPA at level $2\epsilon$. In the FTPL case, when $|F(p, Z)| = |\langle p, Z\rangle| \leq \|Z\|_\infty$, we have

$$\mathbb{E}\left[\max_p F(p, Z) - \min_p F(p, Z)\right] \leq 2\mathbb{E}\|Z\|_\infty \leq \frac{2\log N}{\epsilon}. \tag{16}$$

Fourth, because of the one-step differential stability of the full information GBPA, we begin as in the proof of Lemma C.7 but use a slightly different bound towards the end of the following calculation since we now have $\tilde{q}_{t,i} \geq \rho$.

$$\mathbb{E}\left[\langle p_t - p_{t+1}, \phi_t(\hat{\ell}_t)\rangle\right] = \mathbb{E}\left[\sum_{i:E_{i,t}=j_t}(p_{t,i}-p_{t+1,i})\hat{\ell}_{t,j_t}\right] \leq \mathbb{E}\left[\sum_{i:E_{i,t}=j_t}(2\epsilon\hat{\ell}_{t,j_t}p_{t,i})\hat{\ell}_{t,j_t}\right]$$

$$= \mathbb{E}\left[2\epsilon\hat{\ell}_{t,j_t}^2\sum_{i:E_{i,t}=j_t}p_{t,i}\right] = \mathbb{E}\left[2\epsilon\frac{\ell_{t,j_t}^2}{\tilde{q}_{t,j_t}^2}\tilde{q}_{t,j_t}\right] \leq \frac{2\epsilon}{\rho}\mathbb{E}\left[\ell_{t,j_t}\right]. \qquad (17)$$

Combining (13), (14), (15), (16), and (17) provides

$$\mathbb{E}\left[\sum_{t=1}^{T}\ell_{t,j_t}\right] - L_T^* \leq \left(\frac{2\epsilon}{\rho} + K\rho\right)\mathbb{E}\left[\sum_{t=1}^{T}\ell_{t,j_t}\right] + \frac{2\log N}{\epsilon},$$

where $L_T^* = \min_{i=1}^{N}\sum_{t=1}^{T}\ell_{t,E_{i,t}}$. Denoting the expected regret by $R_T$, we therefore have the bound

$$R_T \leq \left(\frac{2\epsilon}{\rho} + K\rho\right)(R_T + L_T^*) + \frac{2\log N}{\epsilon}.$$

We first set $\rho = \sqrt{2\epsilon/K}$ which is a valid choice as long as $2\epsilon K < 1$. With this choice, we have the bound

$$R_T \leq 2\sqrt{2\epsilon K}(R_T + L_T^*) + \frac{2\log N}{\epsilon}.$$

If $L_T^* \leq 128K\log N$, set $\epsilon = 1/32K$ to get a bound of $R_T \leq L_T^* + 128K\log N \leq 256K\log N$. If $L_T^* > 128K\log N$, set $\epsilon = (\log N)^{2/3}/(K^{1/3}(L_T^*)^{2/3})$ to get a bound of $O(K^{1/3}(\log N)^{2/3}(L_T^*)^{2/3})$. Note that in this case $\epsilon K < 1/32$.

$\square$