[Reviews · NeurIPS 2019]

Reviewer 1



Line 218: missing superscript for Hessian The idea of using algorithmic stability to characterize learnability is a standard one on learning theory and there have been other definitions of stability which have governed online learnability in the following works: Stability Conditions for Online Learnability by Stephane Ross, J. Andrew Bagnell, The Interplay Between Stability and Regret in Online Learning by Ankan Saha, Prateek Jain, and Ambuj Tewari, Policy Regret in Repeated Games by Raman Arora, Michael Dinitz, Teodor V. Marinov, Mehryar Mohri. Maybe it would be beneficial for the paper if the authors briefly discussed the different forms of stability proposed by the above works and how they can be related to Definition 2.1 and Definition 2.2. Overall the paper is clearly written and the intuition and sketches of proofs in the main paper give enough clarity so that the appendix is easy to read. The proofs seem novel and are really a great combination of simple but insightful ideas. I think the attempt at unifying proofs for first order regret bounds for many different types of online learning algorithms is an important one and can give more insights about novel regret minimizing algorithms.

Reviewer 2



This paper reconsidered online learning and bandits from the stability viewpoint by means of differential privacy tools. Though differential privacy is a concept about data security, its main idea is about the output stability, which itself is an important topic in many learning problems including online learning. Back to this paper, authors defined two types of stability based on original \delta approximation max divergence, which were then used to derive first order bounds under full information and partial information feedback respectively. In detail, for online smooth generalized linear loss function, authors obtained first order bound through the first type of stability. As a special case, first order bound for OLO is new. Besides, authors found a close relation between stability defined here with differential consistency, which is also a key concept in the analysis of online learning and bandits. For MAB, authors re-derived some zero order and first order bounds via a unified analysis. Further, a new perturbation based algorithm was proposed to achieve first order bound for bandits with expert advice. Though its theoretical guarantee is not optimal, the algorithm looks interesting and computationally efficient compared with previous optimal but sophisticated algorithm MYGA. The paper is well-written. The idea and analysis in this paper are new, and provide a new approach to analyze general online learning and bandits. Besides, considering some new results are obtained, I tend to accept this paper. However, in my personal view, results in this paper are not so strong in the following sense: 1. First order bounds under full information feedback (mainly Theorem 3.2 and Corollary 3.3) depend on dimension d. Dependency over dimension is a notorious phenomenon in differential privacy literature, so it doesn’t look so surprising that most results in this paper depend on dimension d. However, since first order bound in [1] (see reference below) doesn’t have a such dependency, and we do not need to protect privacy here, I wonder whether it is possible to get rid of the dimensional dependency here; 2. Algorithm 1 is actually computationally inefficient, since we have to solve an ERM at each round, is it possible to provide an elegant update (for example, like perturbed OGD) here? 3. In partial information setting, since we can only calculate the probability p_t approximately via Geometric Resampling, it would be better if authors could derive results under the consideration of approximation error. Overall, I tend to a weak accept about this paper.

Reviewer 3



Originality: The results of this paper unifies and generalizes previous works on this topic. Through drawing connections between online learning and differential privacy, this work makes it possible for scholars in either of these two communities to exploit readily available results in the other community for their research. The paper does a good job comparing and contrasting its results with previous works which highlight the significance of its contributions. Quality: All the claims of the paper are well supported by sound theoretical analyses. This paper is indeed a complete piece of work and it opens up a lot of future research directions to build upon results of this work. I made a high-level check of the proofs in the supplementary file and they seemed to be technically sound. Clarity: This paper is well-written overall. However, I think it's necessary to provide more explanation and intuition for the Obj-Pert and GBPA algorithms in the paper. Currently, it might be a bit hard for someone not familiar with the literature to fully understand the algorithms. Significance: The main contribution of this paper is to draw connections between online learning and differential privacy. Researchers could build upon the results of this work and pursue further research directions in either of these topics. Additionally, the paper provides a unifying framework to analyze a number of different problems that were studied separately in the earlier works.

[Author Response · NeurIPS 2019]

We thank all reviewers for their comments. Minor comments will be addressed in the final version.

## Reviewer 1

**Comparison with related work** Thanks for the references to work of Ross & Bagnell, Saha et al., and Arora et al. All of these papers investigate the relationship between regret and stability of an online learning algorithm and a comparison between different stability conditions certainly makes sense. However, note that these papers do *not* investigate any of the following (that we do in our paper): connections with differential privacy, first order regret bounds, and partial information settings (Arora et al. do consider partial information). We will add a comparison in the final version along with a pointer to different types of stability conditions existing in the literature.

## Reviewer 2

**Comment 1.** Your questioning of the dimension dependence in Theorem 3.2 and Corollary 3.3 is valid. Indeed OGD/FTRL algorithms in these settings will not incur the dimension dependence. However, note that it is not at all clear whether this dimension dependence is due to the use of privacy tools and techniques. Indeed, even the best known *zero-order* bound (ref. [1] in our paper) for OLO in the $\ell_2/\ell_2$ setting via FTPL has a $d^{1/4}$ dependence on dimension $d$. To the best of our knowledge, there is no existing analysis, whether privacy/stability based or otherwise, of *any* FTPL algorithm which does not incur at least this much dependence. It is unknown whether this is an intrinsic limitation of currently available analysis tools or of FTPL methods themselves. In fact, it is somewhat surprising that using the DP based analysis we get first-order regret bounds with a dimension dependence that is the best possible given currently available techniques. Further, this dimension dependence only arises in Theorem 3.2 and Corollary 3.3. Other results in the paper, e.g. experts setting result (Theorem 3.6) and multi-armed bandit result (Theorem 4.2) do not incur any avoidable polynomial dimension dependence.

**Comment 2.** You're right, Algorithm 1 has to solve a convex optimization problem at each step. But it shares this property with FTRL algorithms that have to do the same. As you mentioned, deriving a perturbed OGD type algorithm with a more efficient update will be an interesting topic for future work.

**Comment 3.** The analysis of geometric resampling that takes error due to finite number of samples is already given by Neu & Bartok (ref. [26] in our paper). They showed that error due to drawing $M$ samples contributes an extra $KT/M$ term in the regret where $K$ is the number of arms in a bandit problem. For zero-order $O(\sqrt{T})$ regret bounds, one can therefore choose $M$ of the order of $\sqrt{T}$ (however the expected number of samples needed can be shown to be *constant* per time step, see their Theorem 2). For first-order bounds, one would have to choose a larger $M$ of the order of $T$ which increases the computation. We will add more discussion about this in the final version.

## Reviewer 4

**adding DP definition to the paper** Yes, we can do that.

**rank one Hessian of loss functions** Yes, we are quite positive that the restriction can be removed (personal communication with one of the top experts in DP). In particular, the corresponding DP result should hold for higher ranks at the cost of some degradation in the privacy parameter $\epsilon$. Our online learning result will immediately inherit such a future improvement when it occurs in the DP literature.

**first-order bounds for non-convex problems** This is an intriguing suggestion! The regret bound of the fictitious algorithm $\mathcal{A}^+$ does not use convexity! The only reason convexity is required in Theorem 3.2 is because the DP result of Kifer et al. required convexity. So if one had a DP guarantee for a perturbed ERM algorithm even with non-convex losses, everything would work. Of course, computing the perturbed ERM would involve non-convex optimization so efficient computation may not be possible.

**GPBA for OCO** The best in hindsight action in OLO is $\operatorname{argmin}_{x \in \mathcal{X}} \sum_t \ell_t^\top x = \operatorname{argmin}_{x \in \mathcal{X}} L_T^\top x = \operatorname{argmax}_{x \in \mathcal{X}} (-L_T)^\top x$. This is simply the gradient of the support function of the set $\mathcal{X}$, $y \mapsto \max_{x \in \mathcal{X}} y^\top x$, evaluated at $y = -L_T$. The gradient mapping does not arise in this natural way when the best in hindsight action $\operatorname{argmin}_{x \in \mathcal{X}} \sum_t f_t(x)$ is considered in the case of convex functions $f_t$.

**DC assumption seems more applicable to the bandit setting** A clarification is needed here: DC assumption was indeed introduced by Abernethy et al. (ref. [2] in our paper) in the bandit setting but note that it is actually an assumption on the *potential function*. If you look at the (short!) proof of Theorem 3.6 in Appendix B.3, you will see that DC property of a set of potentials also leads to regret bounds in the full-information experts setting. To summarize, the DC assumption is an assumption on potential functions. Certain potential functions (e.g., those that satisfy DC with a larger exponent $\gamma$) are better suited for bandit settings but the assumption itself is not at all tied to the bandit setting.

[Meta-Review · NeurIPS 2019]

The reviewers reach a consensus that this work provides solid results in connecting online learning and differential privacy. Please do incorporate reviewers' suggestions to the final version.